# The Histone Variant MacroH2A1 Regulates Key Genes for Myogenic Cell Fusion in a Splice-Isoform Dependent Manner

**DOI:** 10.3390/cells9051109

**Published:** 2020-04-30

**Authors:** Sarah Hurtado-Bagès, Melanija Posavec Marjanovic, Vanesa Valero, Roberto Malinverni, David Corujo, Philippe Bouvet, Anne-Claire Lavigne, Kerstin Bystricky, Marcus Buschbeck

**Affiliations:** 1Cancer and Leukemia Epigenetics and Biology Program, Josep Carreras Leukaemia Research Institute (IJC), Campus ICO-GTP-UAB, 08916 Badalona, Spain; shb.scientific.illu@gmail.com (S.H.-B.); vvalero@carrerasresearch.org (V.V.); rmalinverni@carrerasresearch.org (R.M.); dcorujo@carrerasresearch.org (D.C.); 2Program for Predictive and Personalized Medicine of Cancer, Germans Trias i Pujol Research Institute (PMPPC-IGTP), 08916 Badalona, Spain; melanija.posavec@gmail.com; 3Université de Lyon, Ecole Normale Supérieure de Lyon, Centre Léon Bérard, Centre de Recherche en Cancérologie de Lyon, INSERM 1052, CNRS 5286, F-69008 Lyon, France; philippe.bouvet@ens-lyon.fr; 4Center for Integrative Biology (CBI), LBME, University of Toulouse, UPS, CNRS, F-31062 Toulouse, France; anne-claire.lavigne@ibcg.biotoul.fr (A.-C.L.); kerstin.bystricky@ibcg.biotoul.fr (K.B.)

**Keywords:** histone variants, myogenic differentiation, myotubes, cell fusion, macroH2A, PARP1, gene regulation, ADP ribose

## Abstract

MacroH2A histone variants have functions in differentiation, somatic cell reprogramming and cancer. However, at present, it is not clear how macroH2As affect gene regulation to exert these functions. We have parted from the initial observation that loss of total macroH2A1 led to a change in the morphology of murine myotubes differentiated ex vivo. The fusion of myoblasts to myotubes is a key process in embryonic myogenesis and highly relevant for muscle regeneration after acute or chronic injury. We have focused on this physiological process, to investigate the functions of the two splice isoforms of macroH2A1. Individual perturbation of the two isoforms in myotubes forming in vitro from myogenic C2C12 cells showed an opposing phenotype, with macroH2A1.1 enhancing, and macroH2A1.2 reducing, fusion. Differential regulation of a subset of fusion-related genes encoding components of the extracellular matrix and cell surface receptors for adhesion correlated with these phenotypes. We describe, for the first time, splice isoform-specific phenotypes for the histone variant macroH2A1 in a physiologic process and provide evidence for a novel underlying molecular mechanism of gene regulation.

## 1. Introduction

Histone variants can replace replication-coupled histones in the nucleosome and thus convey unique properties to specific regions of the genome [1]. MacroH2A is a member of the group of H2A histone variants and is unique in having a tripartite domain structure consisting of a histone fold, an unstructured linker and a globular macrodomain [2]. Two genes, H2AFY and H2AFY2, and one event of alternative splicing give rise to three proteins sharing the same overall domain structure: the splice variants macroH2A1.1 and macroH2A1.2 and the second gene product macroH2A2 [3]. A number of loss-of-function studies concluded that macroH2As stabilize the epigenome of differentiated cells. In this context, macroH2As were found to provide robustness to differentiation and developmental processes [4,5]. Conversely, macroH2As act as barriers to somatic cell reprogramming [6,7,8]. In cancer, macroH2As primarily have tumor-suppressive functions, which were first described in melanoma [9]. Some isoform-related and cancer-type-related dependences have also been reported [10]. At the present, it is not clear how exactly macroH2As exert their impact on differentiation, cancer and reprogramming. The hypothesis put forward in most of these studies is that the physiological impact of macroH2As is a consequence of downstream changes in gene regulation. This may be a direct consequence of modulating transcription factor binding to DNA [11,12,13] or an indirect consequence of changing the three-dimensional organization of chromatin in the nucleus [14].

The difference between the two splice-isoforms, macroH2A1.1 and macroH2A1.2, is the usage of a mutually exclusive exon that encodes a number of amino acids defining the form and hydrophobicity of a cavity in the macrodomain [15]. As a consequence, macroH2A1.1, but not macroH2A1.2, is able to bind NAD+-derived ADP-ribose, poly-ADP-ribose and ADP-ribosylated PARP1 [16]. In particular, the interaction with the key stress sensor PARP1 can have different consequences. The recruitment of active PARP1 to macroH2A1.1-bound genes can positively or negatively affect their expression [17]. Furthermore, in cases where macroH2A1.1 is more abundant than PARP1, such as in muscle cells, macroH2A1.1 acts as endogenous inhibitor of PARP1 [18,19]. As PARP1 is an avid consumer of nuclear NAD+, its inhibition can have a global impact on NAD+ metabolism affecting reactions in distal organelles, such as oxidative phosphorylation in the mitochondria (discussed in [20]).

The expression of both macroH2A1 splice isoforms is dynamically regulated during myogenesis. The early steps of myogenesis can be recapitulated ex vivo. Primary myoblasts or immortal myogenic cell lines proliferate rapidly in cell culture, but after a change to serum-restricted medium, they quickly exit the cell cycle, upregulate differentiation, promoting transcription factors such as MyoD and Myogenin, aggregate and fuse to form multinucleated myotubes [21]. MacroH2A1.2 is the predominantly expressed isoform in proliferating myoblasts and contributes to the activation of enhancers by recruiting the transcription factor Pbx1 at the beginning of the differentiation process [13]. During myogenesis, a switch in splicing leads to the downregulation of the macroH2A1.2 transcript and an increase in the macroH2A1.1 transcript, yielding comparable protein levels in myogenic C2C12 cells after four days of differentiation [19].

How macroH2As and, in particular, the two splice isoforms, macroH2A1.1 and macroHA1.2, influence physiological processes is not well understood. Here, we have used differentiating myotubes as a model system to examine the role of macroH2A1 splice variants in a differentiation process. We find that the two isoforms have an opposing effect on myotube fusion. The effect is mediated by the regulation of a number of genes encoding adhesion and extracellular matrix proteins.

## 2. Materials and Methods

### 2.1. Cell Culture and Isolation of Primary Myoblasts

Murine C2C12 cells were obtained from ATCC (CRL-1772) and cultured and differentiated as described [19]. Muscles were collected from 2-month-old macroH2A1 knockout mice and control wild-type littermates with a mixed C57Bl/6 and 129Ola background [22]. These experiments were approved by the Ethics Committee for Animal Experimentation of the Ecole Normale Supérieure de Lyon (CECCAPP) under authorization from the National Committee for Ethical Reflection on Animal Experimentation (authorization N°15) on February 22, 2012 with the registered number ENS_2012_008. Primary myoblasts were isolated, cultured and differentiated, as described elsewhere [23]. In brief, mouse primary myoblasts were maintained on collagen-coated dishes in Ham’s F10 medium complemented with 20% FBS (Invitrogen), 10 ng/mL of bFGF (Invitrogen), 0.1% Fungizone (Invitrogen) and penicillin– streptomycin. For maintenance, mouse primary myoblasts were seeded on plates coated with rat-tail collagen I (BD Biosciences). For differentiation, cells were cultured on plates coated with Matrigel™ (BD Biosciences).

### 2.2. Plasmids, Transfection and Retroviral Infection

For the knockdown of macroH2A1 isoforms, we used siRNAs (Invitrogen) that were previously described [19,24]. For all siRNAs, 10 nM was found to be the optimal concentration to achieve a maximal depletion. Briefly, in a 6-well plate, 20,000 C2C12 cells were seeded. C2C12 cells were then transfected with the transfection reaction composed of 500 μL of Opti-MEM (Gibco) 3.12 μL of lipofectamine RNAiMAX Transfection Reagent (Invitrogen) and 10 nM of siRNA. For knockdown in myotubes, siRNA was repeatedly delivered at day +1 after seeding, the day of medium change for differentiation and 2 days after. In the case of p15 plate, 300,000 C2C12 cells were seeded and transfected with 7.5 mL of Opti-MEM, 45 uL of lipofectamine and siRNA at a final concentration of 10 nM. For the reintroduction of Flag-macroH2A1.1 plasmid into primary myoblasts, we generated a retroviral vector, using standard cloning techniques. The mouse macroH2A1.1 was amplified from C2C12 cell cDNA and inserted into a retroviral pBabe.puro backbone. Infection was essentially performed as previously described [25]. Sequences of oligos used for cloning are given in Table A1 in Appendix A.

### 2.3. Antibodies and Immunofluorescence

Please find a description of all used antibodies in Table A2 in Appendix A. The macroH2A1.1 antibody has been generated in the laboratory of Prof. Kerstin Bystricky and is characterized in [26]

For immunofluorescence, 20,000 C2C12 cells were seeded in 6 well-plates and cover slips. On day 4 of differentiation, C2C12 cells were fixed in 4% paraformaldehyde for 10 min at room temperature. Cells were permeabilized for 10 min, using a solution of 0.1 M HCl and 0.5% Triton- × 100 in PBS, and washed three times with PBS containing 0.1% Tween 20 (PBST). Then, slides were pre-blocked with PBST containing 5% BSA, for 30 min, at room temperature, and incubated with a 1/50 to 1/100 dilution of specific primary antibody for 2 h, in PBST containing 5% BSA. After three washes with PBST, the slides were incubated for 1 h, at room temperature, with a 1/100 dilution of secondary goat anti-rabbit antibody conjugated to Alexa 488 (Thermo Fischer). After successive washes with PBST, slides were mounted with Vectashield^®^ Mounting Medium with DAPI.

### 2.4. Image Data Analysis

Images of the C2C12 immunofluorescence were obtained by using a Leica DMI6000B Advance Fluorescence microscope (Leica) microscope equipped with a 63 × /1.4 Plan-Apochromat oil immersion objective. Groups of 10 to 20 images were loaded and analyzed in Fiji, a distribution of ImageJ [27]. Myotubes were directly drawn as Regions of Interest (ROIs) on the selected image, using Fiji’s manual selection tools on the basis of eMHC staining. The images were then automatically analyzed by using a custom ImageJ script, with the aim of calculating the number of nuclei inside each myotube, the total number of nuclei in each image and measurements of each myotube. In short, after background removal with a rolling ball algorithm, the DAPI signal was automatically thresholded, and artifacts were removed with binary image functions and subject to the “analyze particles” function, to identify and count all nuclei in the image. Then, the centroid of each nucleus was extracted, and each region defined as eMHC positive myotubes was evaluated to count the number of nuclei it contains. The total number of nuclei was normalized to the total imaged surface. The differentiation index was calculated as the ratio of nuclei in eMHC-positive cells over the total nuclei. Nuclei distribution per myotube was analyzed by the distribution of nuclei in eMHC-positive cells that contain at least 2 nuclei. The myotube distribution shows different categories of myotubes containing the selected number of nuclei. Total nuclei number and differentiation index were statistically analyzed with Student’s *t*-test, and we used Wilcoxon test for fusion parameters that were not normally distributed.

### 2.5. Gene Expression Analysis

RNA was extracted by fusing the PureLinkTM RNA Mini Kit (Ambion, Life Technologies), including on-column PureLink DNase treatment. RNA concentration was checked by using Nanodrop (ThermoScientific). For RNA-seq, RNA was sent for library preparation to the Genomics Unit of the Centre for Genomic Regulation (CRG, Barcelona, Spain). RNA quality control showed the RNA integrity number (RIN) was between 9.9 and 10. RNA was amplified by using mRNAseq, which selects polyA. The sequencing was performed on single read (SR), with an average length of reads of 50 bp, on an Illumina HiSeq2500 sequencer. RNAseq analysis was performed by using a pipeline assembled in our laboratory. The quantification of the expression was performed by using Salmon software (version 0.13.1) [28] with the quasi-mapping mode on Mus_musculus GRCm38 assembly genome. The statistical analysis was performed by using DESeq2 package from Bioconductor. The raw quantification data obtained with Salmon was imported in DSEq2, using tximport package. Then, we performed three different contrasts, to evaluate the difference in the expression between wild type, si macroH2A1.1 and si macroH2A1.2. The results of this test were corrected, using Benjamini Hochberg procedure (False Discovery Rate). Finally, data of the different contrasts were selected, using an adjusted *p*-value equal or less of 0.05. Gene ontology analysis was performed based on the annotated mouse genome from org.Mm.eg.db (Bioconductor). Pathways analysis was obtained, using Paintomics3 [29]. For RT-qPCR, Complementary DNA (cDNA) was synthesized from 1 μg of RNA by using a first-strand cDNA synthesis kit (Fermentas). Semiquantitative cDNA levels were determined by real-time PCR. Relative expression of genes of interest was determined by normalizing the levels of corresponding cDNAs to the average of the two housekeeping genes Gapdh and Rpl7. All oligos were purchased from Invitrogen (sequences are given in Table A1).

### 2.6. Chromatin immunoprecipitation, Sequencing and Data Analysis

Chromatin immunoprecipitation was performed, essentially, as described elsewhere [30]. For each immunoprecipitation, 25 μg of chromatin and 1–3 μg of antibody were used. Final concentration was determined by QubitTM dsDNA HS Assay (Invitrogen). Precipitated DNA was analyzed by ChIP-qPCR, and the data were normalized on the diluted input (1/50). For ChIP-sequencing (ChIP-seq), 10 ng of enriched DNA was used for library generation and parallel sequencing on Illumina Genome Analyzer at the Genomics Unit of the CRG (Barcelona, Spain). After data cleaning and trimming, reads were aligned to the mouse genome (mm9), using Bowtie 2 version 2.0.6, with sensitive presetting option (-D 15 -R 2-L 22 -i S,1,1.15) [31]. Enriched genomic regions for multiple overlapping ((peaks) SICER software version 1.1) were used to identify enriched regions using the following settings: redundancy threshold = 2, window size = 600, fragment size = 250, effective genome fraction = 0.75, gap = 1200 and FDR = 0.05 [32]. 

Datasets: The here described RNA-seq and MacroH2A1.1 ChIP-seq data can be accessed at GEO through the accession number GSE148045. The ChIP-seq data of total macroH2A1 have been previously described [19] and are accessible through GSE78257. The ChIP-seq data of histone H3 K4me2 in myotubes were generated by others and are accessible through GSM918413. 

## 3. Results

### 3.1. Loss of Total MacroH2A1 Reduces the Capacity of Primary Myoblasts to Fuse

Mice lacking both isoforms of macroH2A1 are viable and fertile [22,33]. We took advantage of the availability of these mice to take a first look at a possible function of macroH2A1 in myogenic differentiation. For this, we isolated quadriceps from macroH2A1 knockout (KO) mice and sex-matched littermate controls and verified the loss of expression by immunoblotting (Figure 1A). Similar to the original studies describing the KO mice, we did not observe any compensation by macroH2A2. Previous quantification of the relative mRNA levels in myotubes indicated that macroH2A2 is expressed at a much lower level than macroH2A1 [19]. In order to investigate the physiological relevance of macroH2A1 in muscle maturation, we isolated primary myoblasts from macroH2A1 wild-type control and knockout (KO) mice and assessed their capacity to differentiate into myotubes ex vivo. As shown in a representative experiment in Figure 1B, although both WT and KO myoblast were able to differentiate and expressed the differentiation marker embryonic myosin heavy chain, the resulting myotubes displayed morphological differences. The lack of large myotubes in macroH2A1 KO conditions was particularly obvious. As cell fusion is a direct function of cell density, we first verified that the number of nuclei was in a similar range in both conditions (Figure 1C). Next, we assessed commitment to differentiation by counting the number of nuclei in in mono or poly nucleated cells expressing embryonic myosin heavy chain (eMHC) as marker of differentiation. We found that the commitment to differentiation was conserved and not affected in KO cells. However, when counting the number of nuclei in poly-nucleated cells, we found that fusion events were strongly reduced in the absence of macroH2A1. A more detailed analysis of the size distribution of the myotubes in all experiments, together, indicated a reduction of the median number of nuclei per myotube. In particular, fewer macroH2A1 KO myotubes with a range of 4–16 nuclei were found, while the number of myotubes with three or two nuclei and cells with a single nucleus were increased (Figure 1D). Re-expression of exogenous macroH2A1.1 in macroH2A1 KO myoblasts led to a visible increase in the formation of larger myotubes (Figure 1E). Taken together, these results indicate that macroH2A1 promotes the fusion of myoblasts to myotubes.

### 3.2. MacroH2A1 Isoforms Have an Opposing Effect on Cell Fusion—MacroH2A1.1 Promotes and MacroH2A1.2 Reduces It

In order to investigate the relative contributions of the macroH2A1.1 and macroH2A1.2 splice isoforms to fusion, we decided to switch to immortal C2C12 cells that recapitulate the myogenic differentiation process in a robust manner. As the time-point of analysis, we chose four days after induction of differentiation, when both isoforms reached a comparable level of protein expression in our hands [19]. We used previously validated siRNAs to interfere with the expression of two isoforms and confirmed the specificity of the interference by immunoblotting (Figure 2A). We have previously shown that, under these conditions, the upregulation of the early differentiation markers *MyoD1* and *Myogenin* is not affected [19]. We observed that the individual knockdowns of both macroH2A1.1 and macroH2A1.2 affected the morphology of myotubes, but in a clearly distinct manner (Figure 2B). Staining for eMHC revealed the absence of extended myotubes in macroH2A1.1 knockdown cells. In contrast, myotubes lacking macroH2A1.2 showed the opposite trend. MacroH2A1.2-deficient myotubes were well organized and in parallel orientation, while macroH2A1.1-deficient myotubes were less organized and more randomly oriented compared to the control. Total nuclei numbers were not affected by either RNA interference; however, knockdown of macroH2A1.2 specifically caused an increase in the percentage of differentiated MHC-expressing cells (Figure 2C). This was also reflected in an increase of total eMHC protein levels detected by immunoblotting (Figure 2A). Counting the number of nuclei per poly-nucleated myotube indicated that overall fusion was decreased in cells knocked-down for macroH2A1.1 and increased in cells knocked-down for macroH2A1.2 (Figure 2D). A more detailed analysis showed that the fraction of smaller myotubes with 2–14 nuclei was increased in macroH2A1.1-depleted conditions, while macroH2A1.2 depletion led to a higher abundance of poly-nucleated myotubes with 15–49 nuclei (Figure 2E). In addition, macroH2A1.2-depleted myoblasts formed particularly large myotubes, with more than 50 nuclei, that were virtually absent under control or macroH2A1.1-depleted conditions. These results suggest that the macroH2A1 splice isoforms affect fusion in an opposite manner. Loss of macroH2A1.1 prevented the formation of myotubes resembling the phenotype of total macroH2A1 knockout (Figure 1), while knockdown of macroH2A1.2 had the opposite effect.

### 3.3. The Opposing Function of MacroH2A1 Isoforms Correlates with the Differential Regulation of a Subset of Genes Related to Extracellular Matrix and Adhesion 

We wondered whether the opposing impact on cell fusion might be caused by changes in gene expression. Therefore, we collected cells after four days of differentiation and sequenced the mRNA. The comparison of macroH2A1 isoform-deficient cells with control cells indicated significant transcriptional changes for the knockdown of both isoforms (Figure 3A). Using an arbitrary cutoff for a log2 fold-change of at least 0.8 corresponding to a fold-change of 1.74 on a linear scale, 995 genes were differentially expressed in macroH2A1.1-depleted cells, and 372 in cells lacking macroH2A1.2. In both cases, the majority of these differentially expressed genes were upregulated, 64% and 62% for si macroH2A1.1 and si macroH2A1.2, respectively. To gain a better understanding of the biological functions of these differentially expressed genes, we separately analyzed enriched gene ontology terms in both upregulated and downregulated genes. Since the depletion of the two isoforms had opposing fusion phenotypes (Figure 2), we paid particular attention to gene ontology terms that would be enriched in sets of differentially regulated genes. We noted that a number of enriched gene ontology terms for biological processes were shared among genes downregulated by si macroH2A1.1 and those upregulated by si macroH2A1.2 (Figure 3B). These included terms relating to migration, adhesion and extracellular matrix organization. The reverse comparison, genes upregulated by si macroH2A1.1 and genes downregulated by si macroH2A1.2, did not yield any shared gene ontology term. 

Direct comparison of differentially expressed in both knockdowns showed an overlap of 98 genes (Figure 3C). The largest group of these genes (46 of 98) was downregulated in macroH2A1.1-deficient cells and upregulated in macroH2A1.2-depleted cells (Figure 3D). Many of these were related to the identified gene ontologies of adhesion, migration and extracellular matrix. We selected five genes and validated their differential expression by RT-qPCR during the course of differentiation (Figure 3E). The chosen genes encoded the ECM-proteins fibronectin, fibromodulin and collagen 1a1 (*Fn1, Fmod* and *Col1a1*, respectively), and the surface proteins integrin alpha 11 and the transmembrane protein 171 (*Itga11* and *Tmem171*, respectively). We were able to confirm the regulation of these genes in an opposing manner by the knockdown of macroH2A1.1 and macroH2A1.2 (Figure 3E). The differential regulation was most pronounced at four days of the differentiation process coinciding with the time-point we observed the fusion phenotype. Taken together our transcriptional analysis suggests that macroH2A1.1 and macroH2A1.2 exert their opposing influence on cell fusion through gene regulation. Our knockdown experiments suggest that macroH2A1.1 promotes, while macroH2A1.2 inhibits, the expression of a number of genes encoding proteins related to adhesion, migration and the organization of extracellular matrix. These genes are likely candidates to mediate the fusion phenotype observed in myotubes lacking specific macroH2A1 isoforms. For simplicity, we herein refer to the group of these genes as fusion genes.

### 3.4. Knockdown of PARP1 Has Little Impact on Fusion and Expression of Related Genes

In our previous study, we reported that macroH2A1.1 impacts NAD+ metabolism by inhibiting PARP1 [19]. Others have reported that PARP1 and macroH2A1 can cooperate in transcriptional regulation [17,18]. Hence, we decided to test whether inhibiting PARP1 would impact myotube fusion. Using previously validated siRNAs against PARP1, we performed experiments in analogous with those shown in Figure 2 and analyzed myotubes after four days of differentiation. Efficient suppression of PARP1 expression was confirmed by immunoblotting (Figure 4A). Staining of myotubes with an antibody for the eMHC marker indicated that both control and PARP1-deficient cells were able to form polynucleated myotubes, without major morphologic differences between both conditions (Figure 4B). In contrast to the knockdown of macroH2A1 isoforms, we did not observe any significant difference between control and PARP1 knockdown when analyzing the distribution of myotubes according to the range of their number of nuclei (Figure 4C). Furthermore, knockdown of PARP1 did not have a major influence on the expression of adhesion and migration genes that were differentially and opposingly regulated by the macroH2A1 isoforms. These results suggest that PARP1 is not a major regulator of fusion and thus lend support to the hypothesis that the impact of macroH2A1 splice isoforms might be independent of PARP1.

### 3.5. MacroH2A1.1 Becomes Enriched on Fusion-Relevant Genes During Differentiation

In order to get a better idea about the genomic distribution of macroH2A1.1, we performed a chromatin immunoprecipitation (ChIP), using an isoform-specific antibody for macroH2A1.1. Sequencing of the precipitated DNA and mapping the reads allowed us to extract information on the genomic distribution of macroH2A1.1 in myotubes. As frequently observed with macroH2A ChIP-seqs [5,6,34,35], the signal-to-noise ratio along the genome was low, suggesting widespread distribution and low focal enrichment. By averaging mapped reads on genes divided in three categories according to their expression, we observed reduced macroH2A1.1 occupancy on the bodies of genes with increasing transcriptional activity (Figure 5A). In this respect, macroH2A1.1 follows the patterns described by others for macroH2A2 and total macroH2A1 [34,36]. It is interesting to note that macroH2A1.1 was enriched before the transcription start site (Figure 5A). We turned back to the list of 46 genes that were regulated in opposing directions by the knockdown of the individual macroH2A1 isoforms and performed a pathway analysis, using the Paintomics3 tool. This highlighted the involvement of many of these genes in interactions of cell surface receptors with proteins of the extracellular matrix and interactions relevant for focal adhesions (Figure 5B). In the context of surface receptors, the presence of several genes encoding integrin subunits was particularly striking. Encoded proteins of the extracellular matrix were fibronectin-1 (*Fn1*), Thrombospondin 1 (*Thbs1*) and collagen type I alpha 1 chain (*Col1a1*). This pathway analysis correlates with the enrichment of terms relating to extracellular matrix organization and adhesion (Figure 3B). The enrichment profile of macroH2A1.1 on these genes was similar to total macroH2A1 in differentiated myotubes and proliferating myoblasts (Figure 5C). Strikingly, RNA sequencing read-count revealed that *Fn1*, *Thbs1* and *Col1a1* were among the ten most expressed genes in C2C12 myotubes. We validated regions of the genes *Fn1* and *Col1a1* and the adhesion-related gene *Itga11* by quantitative PCR of three independent ChIPs (amplicons for *Fn1* and *Col1a1* are indicated Figure 5C). In all cases, macroH2A1.1 levels were increased after four days of differentiation, although the relative levels between loci varied (Figure 5D). Three loci around the *Col1a1* gene showed a clear enrichment for macroH2A1.1, and this signal was lost in macroH2A1.1-depleted cells, but not in macroH2A1.2-depleted cells (Figure 5E).

Taken together, our results suggest that macroH2A1.1 and macroH2A1.2 mediate opposing phenotypes during muscle cell differentiation, by regulating a subset of genes in the opposing direction. The large majority of these genes are downregulated in macroH2A1.1-depleted cells and upregulated in macroH2A1.2-depleted cells. The presence of macroH2A1.1 on or in the proximal vicinity of these genes is suggestive of a direct mechanism favoring gene expression, but most likely independent of PARP1. These genes are related to adhesion, interactions of extracellular matrix proteins with cell surface receptors and migration, and they are likely candidates to explain the influence of macroH2A on cell fusion, which is a key step in myogenesis.

## 4. Discussion

### 4.1. How Can Splice Isoforms of Histone Variant MacroH2A1 Affect Gene Transcription?

The initial observation that macroH2A1 is enriched on the inactive X chromosome suggested a role in gene repression [37]. While the large majority of subsequent studies have confirmed the association of macroH2A with repressed chromatin regions, a few studies reported an involvement in gene activation (discussed in [38]). The common denominator of this second set of studies was that they were looking at gene activation in response to different signals or during dynamic processes, such as the differentiation of stem cells [5,18,35,39]. Here, we report that both the individual knockdown of macroH2A1.1 and macroH2A1.2 deregulate genes during myogenic differentiation. Two-thirds of all deregulated genes in cells deficient for one or the other macroH2A1 splice isoform were upregulated, and this is in line with a suggested primary function in repression. However, the fact that one-third of genes were downregulated indicated a more complex role of macroH2A1 in transcriptional regulation. The large majority of deregulated genes in the two datasets were non-overlapping. The limited overlap between the genes deregulated by the different macroH2A1 isoforms might be caused by differences in their genomic distribution or expression dynamics. Concerning the latter, we have previously shown that macroH2A1.2 is highly expressed in myoblasts and its expression is progressively reduced during myogenic differentiation, while macroH2A1.1 only becomes expressed after cells commit to differentiation [19]. We still lack a good picture of how the two isoforms compare in their genomic distribution. Biochemical evidence suggests that the two macroH2A1 isoforms associate with distinct types of chromatin, in particular macroH2A1.2 with Polycomb-repressed chromatin [17].

Here, we report that opposing phenotypes caused by the separate knockdown of the macroH2A1 isoforms correlate with the deregulation of a subset of genes. As a consequence, we have decided to focus our study on the analysis of these genes and refer to them as fusion genes. The expression of fusion genes was downregulated by the knockdown of macroH2A1.1 and upregulated by the knockdown of macroH2A1.2. This suggests that macroH2A1.1 contributes to the transcription of fusion genes, while macroH2A1.2 would do the opposite. This function might be mediated by direct and indirect mechanisms. A possible explanation for such behavior in a direct mechanism is that macroH2A1.1 would be able to recruit a transcription-promoting factor through its macrodomain’s unique binding pocket and that macroH2A1.2 would be competing with macroH2A1.1 on relevant loci. Indeed, macroH2A1.1 and macroH2A1.2 share the same histone fold domain that contains the docking domain relevant for chromatin incorporation [40]. We were able to detect macroH2A1.1 enrichment on fusion genes (Figure 5), and these levels increased during differentiation, in correlation with the overall increase in macroH2A1.1 expression [19]. Unfortunately, we were unable to test the competition hypothesis, since, in our hands, all tested antibodies failed to specifically detect macroH2A1.2 in chromatin immunoprecipitation (Bystricky and Buschbeck labs), even though we have included antibodies that have been used in previous studies [13]. Several studies have linked the role of macroH2A1.1 in gene regulation to its ability to co-operate with PARP1 [17,18]. On the molecular level, this is thought to happen through the binding of the ADP ribose moiety of auto-ADP-ribosylated PARP1 by the macrodomain of macroH2A1.1 and the PARP1 activity-dependent recruitment of the acetyltransferase CBP to macroH2A1.1-containing loci [17]. Furthermore, we previously reported that macroH2A1.1 can also act as endogenous inhibitor of PARP1 and impact global cellular NAD+ metabolism, in particular when macroH2A1.1 levels are high and exceed the levels of PARP1 [19]. Although very different in nature, PARP1 could mediate macroH2A1.1-dependent gene regulation in both cases. However, we found that, in contrast to macroH2A1 isoforms, interference with PARP1 does not affect the fusion capacity of cells and the expression of genes. Thus, the observed impact on gene regulation cannot only be attributed to an interaction of macroH2A1.1 with PARP1 and a potential competition by macroH2A1.2. An alternative explanation for a direct mechanism would be the recruitment of other transcription-promoting factors by macroH2A1.1. Indeed, the binding mode of macroH2A1.1 with ADP ribose is compatible with the binding of any ADP-ribosylated effector protein if sufficiently abundant [16].

Hypotheses on indirect mechanisms are more difficult to formulate and even more difficult to test in experiments. From our previous studies, we know that macroH2A proteins have a major impact on nuclear organization and three-dimensional chromatin architecture, particularly on heterochromatin [14]. Heterochromatin organization contributes to cell differentiation through channeling gene-expression programs [41]. Although we were able to map the function of macroH2A1 in heterochromatin architecture to the shared linker region [42], we cannot entirely rule out that the two splice isoforms might differentially modulate chromatin architecture in a way that could favor or disfavor the expression of fusion genes. This could be mediated by the isoform-specific recruitment of effector proteins. In this regard, it is interesting to point out that macroH2A1.2 has been suggested to bind the Polycomb protein Ezh2 in a manner that could be isoform-specific [43]. For macroH2A1.1, we have already above discussed its capacity to bind ADP-ribosylated proteins. Taken together, indirect and direct mechanisms could explain the differential and opposing influence of macroH2A1 splice isoforms on differentiation. Currently, we favor a hypothesis on a direct mechanism and will test if macroH2A1.1 can affect gene transcription through the ADP-ribose dependent recruitment of transcriptional regulators other than PARP1.

### 4.2. The Relevance of Identified Genes for Cell Fusion?

Previous studies aiming to dissect the specific cellular functions of the two splice variants of macroH2A1 were conducted in the pathophysiological context of cancer [24,44,45]. We now provide differential analysis of the two splice isoforms in the physiological context of muscle biology. We find that individual knockdowns of macroH2A1.1 and macroH2A1.2 caused opposing phenotypes during myogenic differentiation. Knockdown of macroH2A1.2 increased myogenic cell fusion, while macroH2A1.1 reduced it (Figure 2). Our observation contrasts with a previous study reporting that knockdown of macroH2A1.2 rendered myoblast unable to differentiate [13]. A major difference in this study was the time-point of the knockdown that was performed before induction of differentiation, while in the current study, we had opted for an interference approach achieving maximal perturbation at four days of differentiation. The regulation of fusion-related genes we observed in our study is incompatible with the previously described transcription-promoting mechanism for macroH2A1.2 that involved the recruitment of the transcription factor Pbx1 to regulatory elements [13]. It is worth noting that the knockdown of macroH2A1.1 in differentiated cells led to a two-to-three-fold higher number of deregulated genes than the knockdown of macroH2A1.2, suggesting a more prominent role for macroH2A1.1 in transcriptional regulation.

Several of the genes identified in Figure 3D to be opposingly regulated by both macroH2A1 isoforms are known to have key roles in myogenic cell fusion. This includes the fibromodulin-encoding *Fmod* and gene *Col1a1* that encodes a collagen subtype. *Fmod* is a positive regulator of *Col1a1*, and its loss reduces fusion of C2C12 cells [46]. Interestingly, in our hands, *Fmod* was the most downregulated gene in cells treated with si macroH2A1.1 (log2 fold change of -4.45) and at the same time the most upregulated in cells treated with si macroH2A1.2 (log2 fold-changes of 4.80). Similarly, knockdown of the cell adhesion receptor-encoding gene *Itga11* inhibits the fusion capacity of muscle satellite cells and myotube formation [47]. The specific role of the fibronectin 1 and the thrombospondin 1 encoded by *Fn1* and *Thbs1*, respectively, in muscle fusion is still unclear. However, both proteins are known to be essential for cell adhesion, cell motility, and cell-to-cell and cell-to-matrix interactions in other cell types [48] and these functions are tightly linked to myogenic cell fusion. Furthermore, it is interesting to point out that fibromodulin and fibronectin proteins have been predicted to directly interact with each other and to have antagonistic functions in myogenesis [49]. In light of this, it is interesting to point out that the downregulation *Fmod* during differentiation was lost in cells lacking macroH2A1.2 (Figure 3E). This leads us to speculate that macroH2A1 isoforms do not only regulate a few key genes involved in muscle fusion but also support transcription factors in orchestrating entire expression programs essential for the proper timing of fusion initiation and myotube maintenance. 

## 5. Conclusions

In conclusion, here we describe an isoform-specific function of macroH2A1 splice isoforms in myogenic cell fusion that we link to the regulation of fusion-relevant genes encoding extracellular matrix proteins and adhesion proteins. Future work will need to dissect the underlying molecular mechanism and address the relevance of these findings for diseases, such as muscle dystrophies, with defects in the interaction of muscle fiber and extracellular matrix.

## Figures and Tables

**Figure 1 cells-09-01109-f001:**
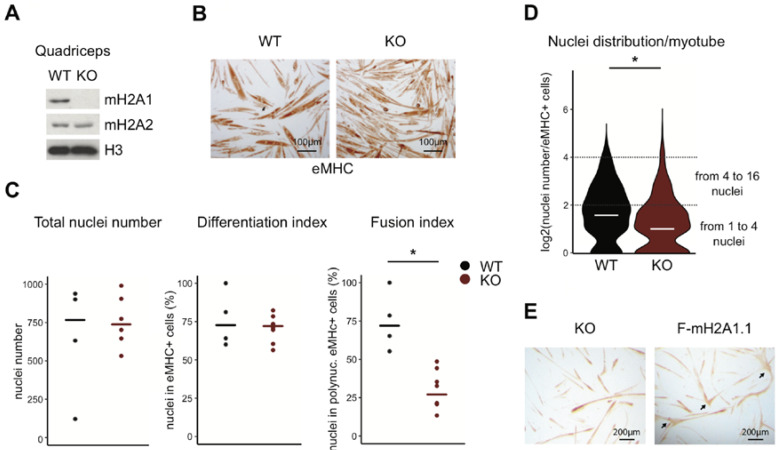
Myoblasts lacking macroH2A1 fail to form proper myotubes. (**A**) Immunoblot analysis of macroH2A1 (mH2A1), mH2A2 and Histone H3 of quadriceps from wild-type (WT) and knockout (KO) mice. (**B**) Immunostaining of differentiated primary mH2A1 WT and KO myotubes with anti-embryonic myosin heavy chain (eMHC). (**C**) Total nuclei, percent of differentiated eMHC positive cells and percent of differentiated eMHC positive cells with at least two nuclei after four days of differentiation. Data points are from sets of four to six photos obtained from three independent biological replicates, evaluating 100 and 200 myotubes and the same surface area per replicate. Lines indicate the median of replicates, * *p* < 0.05, Student’s *t*-test. (**D**) Violin graph of the nuclei distribution per primary mH2A1 WT and KO myoblasts. The data shown are the log2 of the median of 100–200 myotubes from three biological replicates, * *p* < 0.05, Wilcoxon test. (**E**) Bright field pictures of anti-eMHC immunostainings of primary macroH2A1 KO myoblasts transduced with Flag-mH2A1.1 or control vector after two days of differentiation. Arrows indicate large myotubes.

**Figure 2 cells-09-01109-f002:**
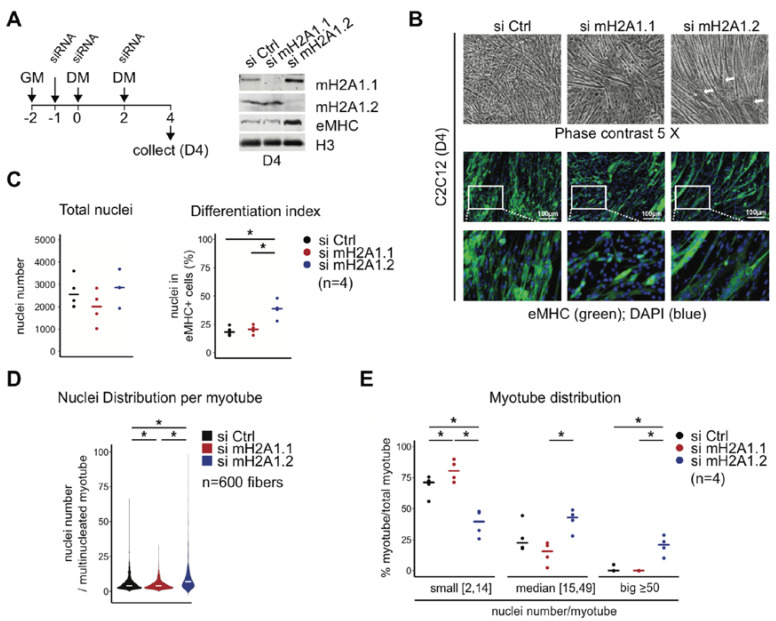
MacroH2A1 isoforms oppositely regulate myotube fusion. (**A**) A schematic representation of the used RNA interference protocol, and the resulting protein levels in C2C12 cells are shown. Immunoblotting was performed by using indicated antibodies. Differentiation was induced by changing growth medium (GM) to differentiation medium (DM), and samples were collected after four days (D4). (**B**) Differences in C2C12 myotube morphology are visible in phase contrast and by anti-eMHC immunofluorescence at D4. Nuclear DNA was counter-stained by DAPI. White arrows indicate particularly large myotubes. (**C**) The total nuclei number and the percentage of differentiated eMHC-positive cells were assessed at D4. Same areas with 600 myotubes were analyzed; data points are from four areas obtained from two independent biological replicates, * *p* < 0.05; Student’s *t*-test). (**D**) Violin graph of the nuclei distribution per myotubes. Data in D are the median of 600 myotubes from two biological replicates, * *p* < 0.05, Wilcoxon test. (**E**) Percent of myotube distribution between three groups: myotubes containing between 2 and 14, between 15 and 49, and more than 50 nuclei. Data points are the median of 600 myotubes, from four areas obtained from two independent biological replicates, * *p* < 0.05, Student’s *t*-test.

**Figure 3 cells-09-01109-f003:**
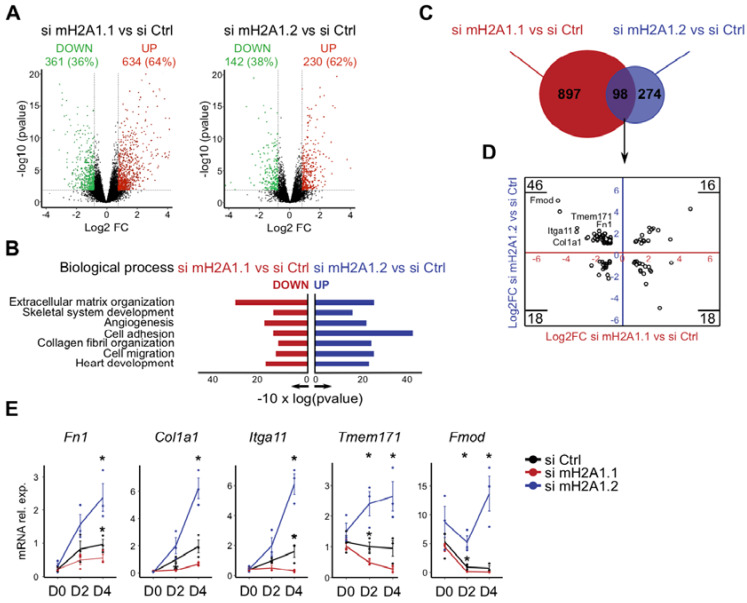
MacroH2A1 (mH2A1) isoforms affect genes related to extracellular matrix and adhesion in an opposing manner. (**A**) Volcano plot of -log10 of the adjusted *p*-value versus the log2 fold-change (FC). Dashed lines demarcate the chosen cutoffs of <0,01 and 0.8 for *p*-value and log2FC, respectively. Significant deregulated genes are highlighted in green (down) or orange (up). The analyzed data were obtained from three independent biological replicates. (**B**) Gene ontology analysis of downregulated genes with si mH2A1.1 versus upregulated with si mH2A1.2 (*p* < 0.05). The analyzed data were obtained from three independent biological replicates. (**C**) Venn diagram of deregulated genes identified in (**A**). (**D**) Scatterplot of deregulated genes (98) identified in (**A**) and shared in both comparisons, si mH2A1.1 versus control and si mH2A1.2. Five fusion genes of interest are highlighted. (**E**) Validation of selected genes by RT-qPCR during differentiation at days 0, 2 and 4 (left to right). Data is the mean of four independent experiments + SD; * *p* < 0.05; Student’s *t*-test.

**Figure 4 cells-09-01109-f004:**
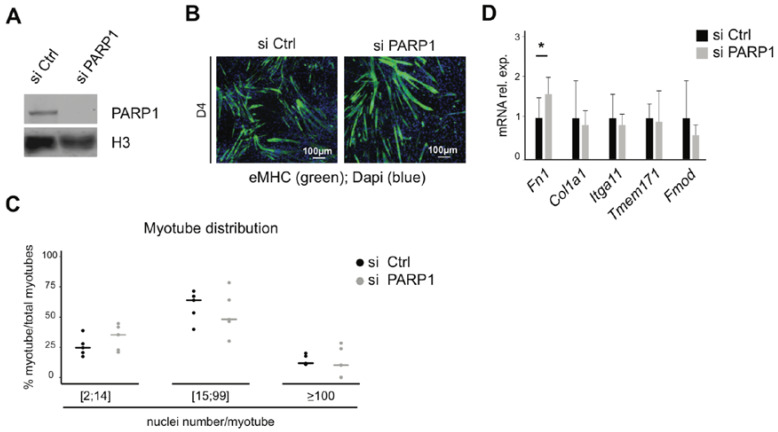
Silencing of PARP1 does not affect cell fusion or expression of fusion genes. (**A**) Successful PARP1 silencing by siRNA is shown by immunoblotting of C2C12 cells after four days of differentiation. (**B**) Myotubes were analyzed by eMHC immunofluorescence in si Ctrl versus si PARP1 conditions at day 4. Nuclear DNA was stained by DAPI. (**C**) Percentage of myotube distribution divided into three groups, according to the range of nuclei per myotube. Data in C are the median of 600 myotubes from four distinct areas from three biological replicates, * *p* < 0.05, Student’s *t*-test. (**D**) Relative mRNA levels of five fusion genes (identified in Figure 3) in si Ctrl and si PARP1 conditions. Data in D are the mean of four independent experiments + SD; * *p* < 0.05; Student’s *t*-test.

**Figure 5 cells-09-01109-f005:**
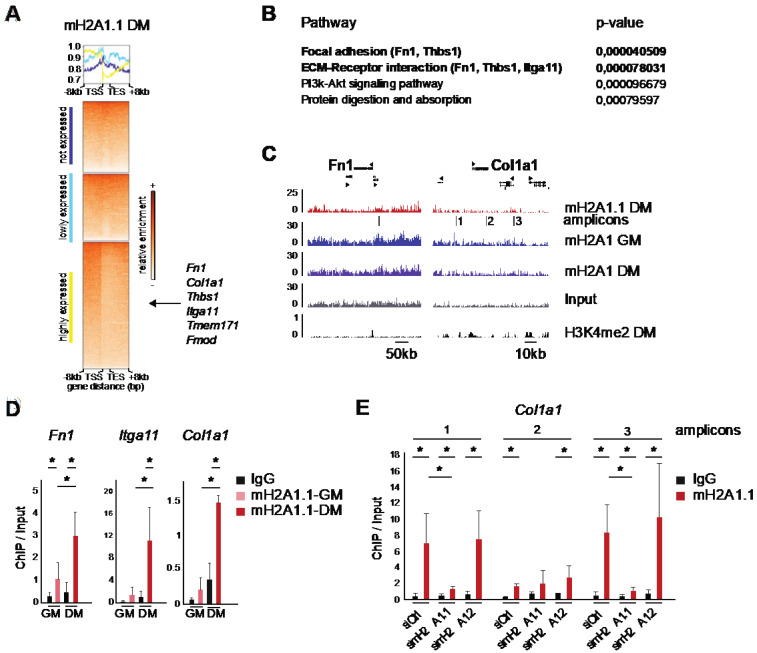
MacroH2A1.1 (mH2A1.1) occupies fusion-related genes in myotubes. (**A**) Heatmaps and summary plots using normalized read coverages from mH2A1.1 ChIP-seq from myotubes after four days of differentiation. Signals over the body of all genes +/- 8 kb are shown. Heatmaps were divided into three categories, according to normalized read counts: not expressed, lowly expressed and highly expressed genes. Fusion genes of interest belong to the category of highly expressed genes. (**B**) Pathway analysis of the 46 genes of interest by Paintomics3 analysis. (**C**) Snapshots from the UCSC genome for macroH2A1.1, total macroH2A1 and the enhancer mark histone H3K4me2 in proliferating myoblasts in growth medium (GM) or myotubes in differentiation medium (DM) and input are shown. The exact coordinates are indicated in Table A1. (**D**) Levels of macroH2A1.1 enrichment by ChIP-qPCR on *Fn1*, *Itga11* and *Col1a1* in proliferative (GM) and differentiated (DM, four days) cells. As a background control, we used IgG. Data are mean of *n* = 3 independent experiments, normalized to the input signal, + SD; * *p* < 0.05; Student’s *t*-test. (**E**) Levels of macroH2A1.1 enrichment by ChIP-qPCR on three loci of *Col1a1* genes, in si ctrl, si macroH2A1.1 and si macroH2A1.2 conditions at day 4. As a background control, we used IgG. Data are the mean of n = 4 independent experiments, normalized to the input signal, + SD; * *p* < 0.05; Student’s *t*-test.

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
