# Peer review of "The Histone Variant MacroH2A1 Regulates Key Genes for Myogenic Cell Fusion in a Splice-Isoform Dependent Manner"

_cells, 2020, doi:10.3390/cells9051109_

Round 1

Reviewer 1 Report

This article by Hurtado-Bagès and colleagues describes isoform-specific phenotypes for the histone variant macroH2A1 in a physiologic process for a novel underlying molecular mechanism of gene regulation. Authors have used differentiating myotubes as a model system to examine the role of macroH2A1 splice variants in a differentiation process and the two isoforms have an opposing effect on myotube fusion. The effect is mediated by the regulation of several genes encoding adhesion and extracellular matrix proteins. The study has potentially interesting new data, but the manuscript requires improvement in organization and clarity.

  1. Fig 1B, shows WT and KO myoblast displayed morphological differences. The figure doesn’t have any controls or scale, it makes it difficult to compare them together. Authors should also add interpretation for the differences.
  2. Figure 1, legend say 100-200 myotubes were analyzed, are the number not consistent among different groups? Please specify the number for each group.
  3. Figure 1E- staining with eMHC like in figure 1B will be more helpful in confidence of rescue. 1B and 1E can include both eMHC staining and Bright field for consistency in figures.
  4. Figure 1, Quantification of differences in large myotubes KO vs WT will be helpful.
  5. Quantification on amount of macro H2A1.1 in WT vs KO will be helpful. Also comments on amount of rescue will make data more comprehensible.
  6. Authors should comment why the RNA seq gene ontology top pathway are not similar to the chip seq data. Are there other factors involve for gene regulation via macroH2A1.1?
  7. Page4, line187 Rephrase the sentence for understanding- Judged by the number of nuclei in differentiated cells marked by the expression of embryonic myosin heavy chain (eMHC) cells, we further found that also the commitment to differentiation was conserved (also not affected).
  8. Line 199- needs English editing
  9. Figure 2, authors should add GM (growth media) and DM (differentiation media) in legend.

Author Response

We would like to thank this reviewer for having taking their time to carefully revise our manuscript and to provide valuable comments. We were glad that the reviewer considered our study to present ‘interesting new data’. We have addressed the comments and hope the reviewer will find our revised version to have an adequate level of clarity and improved organization.

Comment 1: Fig 1B, shows WT and KO myoblast displayed morphological differences. The figure doesn’t have any controls or scale, it makes it difficult to compare them together. Authors should also add interpretation for the differences.

Our answer: Indeed, our figures missed scale. We apologize for that the oversight. We have now added the scale in the revised Figures 1B and 1E. We now present our interpretation of the observations in the concluding sentence of the paragraph (line 194): Taken together, these results indicate that macroH2A1 promotes the fusion of myoblasts to myotubes.

Comment 2: Figure 1, legend say 100-200 myotubes were analyzed, are the number not consistent among different groups? Please specify the number for each group.

Our answer: We apologize if the legend was not clear enough. In 1C, we compared the same surface area between different replicates and conditions. Depending on the degree of fusion these areas contained between 100 and 200 fibers. In Figure 1D, data from three biological replicates has been combined. and in total of Figure 1C is shown combined. We changed the legend of the figures to clarify this matter (line 199-205).

Comment 3: Figure 1E- staining with eMHC like in figure 1B will be more helpful in confidence of rescue. 1B and 1E can include both eMHC staining and Bright field for consistency in figures.

Our answer: Following the suggestion of the reviewer, we have exchanged the bright field image for eMHC staining (Figure 1E).

Comment 4: Figure 1, Quantification of differences in large myotubes KO vs WT will be helpful.

Our answer: The quantification was performed on eMHC-positive cells and myotubes. The difference between both conditions was particular obvious for myotubes containing between 4 and 16 nuclei. As the log scale is not easy to read we have indicated these area in Figure 1D and emphasized the observation in the text (line 190).

Comment 5: Quantification on amount of macroH2A1.1 in WT vs KO will be helpful. Also comments on amount of rescue will make data more comprehensible.

Our answer: We fully agree with the reviewer. The amount of infected myoblasts that we obtained was very limiting and while enough for the analysis of the fusion phenotype we did not obtain enough to assess the amount of the exogenous protein. When infecting other cells with the same vector, we usually achieved levels that were close to the endogenous levels of macroH2A1.1 in C2C12 cells.

Comment 6: Authors should comment why the RNA seq gene ontology top pathway are not similar to the chip seq data. Are there other factors involve for gene regulation via macroH2A1.1?

Our answer: We apologize if our data representation was misleading. The pathway analysis was performed on 46 genes that were identified by RNA-seq to be down regulated by the knockdown of macroH2A1.1 and up regulated by the knockdown of macroH2A1.2 (Figure 3D). This is now clarified at the beginning of a new paragraph (line 341). Paintomics 3 is a tool to identify (and visualize) enriched KEGG pathways with smaller gene sets and we felt it nicely complemented the gene ontology analysis shown in Figure 3.

Comment 7: Page4, line187 Rephrase the sentence for understanding- Judged by the number of nuclei in differentiated cells marked by the expression of embryonic myosin heavy chain (eMHC) cells, we further found that also the commitment to differentiation was conserved (also not affected).

Our answer: We have now rephrased in two sentences to make this more clear (now line 184).

Comment 8: Line 199- needs English editing.

Our answer: The sentence has been edited (now line 194).

Comment 9: Figure 2, authors should add GM (growth media) and DM (differentiation media) in legend.

Our answer: We apologize for this omission. We have now added this information in the legend (line 242).

Once again we would like to thank Reviewer 1 for their suggestions that have helped to further improve the manuscript.

Reviewer 2 Report

  1. I enjoyed reading this article, for its clarity and interesting work.
  2. In the Material and Methods section, they mention several times that the NGS services were performed at the CRG Genomics Unit, but there is no information of where it is located (a University or Research Institute or a company?). This information needs to be included.
  3. On page 3, lines 142 and 143, it is indicated that one of the co-authors performed the statistical analysis. This sentence is unnecessary and should be eliminated.
  4. In the Materials and Methods section, no information is provided on number of biological replicates used in each type of experiment (mice, cell lines), some of that information is indicated in parts of some figures (for example, figure 1D etc) but it is hard to assess data rigor and reproducibility without including the number of replicates in all experimental results or in the methods section.
  5. On page 3, lines 143-144, it is mentioned that the RNASeq data analysis "was performed using a pipeline assembled in our laboratory", and the reference included is for the Salmon software, the pipeline information should be included in the manuscript, since the information provided is too scant and is not sufficient to evaluate data rigor and reproducibility.
  6. There are some minor English errors and typos in the manuscript that need to be corrected
    1. page 3, line 106 - "horse redish" is not correct, should be "horseradish"
    2. page 3, line 107 - the words "the standard" at the end of that line would be eliminated.
    3. Page 3, line 125, the word "artifact" is misspelled, please correct.
    4. Page 3, line 140, mRNA Seq method  (kits) used should be specified. 
    5. page 4, line 159, the word "by" should follow the word analyzed at the end of that line.
    6. Figure 1 title, the word "proper" is missing the first letter "p"
    7. page 8, line 314, the word "in" should be eliminated (where it says "experiments in analogous"
    8. page 10, line 369, Figure 5 legend. The words "to the" should be added where it says "Fusion genes of interest belong (to the) category"
    9. Page 11, line 440 at the end, the word "discard" should be replaced with "rule out"

Author Response

We would like to thank this reviewer for their positive feedback and for suggesting further improvements that we have now performed.

Comment 1: I enjoyed reading this article, for its clarity and interesting work.

Our answer: Thank you very much for your comments and suggestions as well as your interest in our work.

Comment 2: In the Material and Methods section, they mention several times that the NGS services were performed at the CRG Genomics Unit, but there is no information of where it is located (a University or Research Institute or a company?). This information needs to be included.

Our answer: We apologize for the omission and have now included this information (line 133 and line 158). The Center for Genomic Regulation is research institute in Barcelona with excellent service facilities.

Comment 3: On page 3, lines 142 and 143, it is indicated that one of the co-authors performed the statistical analysis. This sentence is unnecessary and should be eliminated.

Our answer: We agree and have removed the sentence as requested.

Comment 4: In the Materials and Methods section, no information is provided on number of biological replicates used in each type of experiment (mice, cell lines), some of that information is indicated in parts of some figures (for example, figure 1D etc) but it is hard to assess data rigor and reproducibility without including the number of replicates in all experimental results or in the methods section.

Our answer: We would like to thank you for this comment. In order to facilitate the understanding of our data, we now provide further information in the legend section for these figures (Figure 1 line 206, Figure 2 line 252 and 257, figure 3 line 283 and 285, Figure 4 line 315). We hope it will help the readers to appreciate the rigor and weight of our results.

Comment 5: On page 3, lines 143-144, it is mentioned that the RNASeq data analysis "was performed using a pipeline assembled in our laboratory", and the reference included is for the Salmon software, the pipeline information should be included in the manuscript, since the information provided is too scant and is not sufficient to evaluate data rigor and reproducibility. 

Our answer: We apologize for lacking detail in the description. We have now added further information related to the pipeline in the material and method section (page 3 line 137 to 144).

Comments 6. There are some minor English errors and typos in the manuscript that need to be corrected.

  1. page 3, line 106 - "horse redish" is not correct, should be "horseradish".
  2. page 3, line 107 - the words "the standard" at the end of that line would be eliminated.
  3. Page 3, line 125, the word "artifact" is misspelled, please correct.
  4. Page 3, line 140, mRNA Seq method (kits) used should be specified. Our
  5. page 4, line 159, the word "by" should follow the word analyzed at the end of that line.
  6. Figure 1 title, the word "proper" is missing the first letter "p"
  7. page 8, line 314, the word "in" should be eliminated (where it says "experiments in analogous"
  8. page 10, line 369, Figure 5 legend. The words "to the" should be added where it says "Fusion genes of interest belong (to the) category"
  9. Page 11, line 440 at the end, the word "discard" should be replaced with "rule out".

Our answer: We would like to thank this reviewer for having spotted all this mistakes and typos. We corrected them as suggested. The information on the used kit was inserted the methods section (line 134). Please not that the antibody information has been moved to Table 2 in Annex A.

Thanks a lot.

Reviewer 3 Report

Dear authors,

you present an interesting study about the differential impact of histone variants H2A1.1 and H2A1.2 on myotube formation. I think this study is of great value as it provides information on the physiological role of these histone variants and not only evaluates their function in the context of various pathophysiological processes like e.g. cancer.

In my opinion their are only minor revisions that need to be addressed before publication of this manuscript.

  1. Did you use isotope controls for your immunofluorescence experiments? You describe automatic analysis of the staining with Fiji. We use the same software during automatic analysis of  cancer tissue sections. Isotype controls are useful in setting up intensity thresholds that optimally separated the isotype control from the active antibody. 
  2. I would recommend to include a table with the used antibodies (and of course isotype controls) in the supplementary material.
  3. Just out of interest for myself, you describe that macroH2A1 mice are viable and fertile. Did you see any differences in muscle function or agility?

All the best for your further publication process.

Author Response

We would like to thank this reviewer for their comments and were glad to read that he found our study of great value. We have now addressed the few remaining comments.

Comment 1: Did you use isotope controls for your immunofluorescence experiments? You describe automatic analysis of the staining with Fiji. We use the same software during automatic analysis of  cancer tissue sections. Isotype controls are useful in setting up intensity thresholds that optimally separated the isotype control from the active antibody.

Our answer: We fully agree with this reviewer on the importance of isotype controls when setting up staining protocols. Indeed, we have set-up the staining conditions with adequate negative controls and found that the background levels were extremely low. Given that the eMHC antibody is well established and widely used in the muscle community for detecting differentiating myotubes, we felt that the inclusion of these control experiments was not necessary. The staining of myotubes but not non-fused cells (see for instance Figure 2B bottom, middle panel) reconfirmed its specificity as differentiation marker.

Comment 2: I would recommend to include a table with the used antibodies (and of course isotype controls) in the supplementary material.

Our answer: We thank the reviewer for the suggestion. We have added such as a table in the annex section (page 14).

Comment 3: Just out of interest for myself, you describe that macroH2A1 mice are viable and fertile. Did you see any differences in muscle function or agility?

Our answer: This is a great question that we are actively pursuing. We have now generated isoform specific knock-out mice that are undergoing phenotyping at the German Mouse Clinic. The standardized phenotyping pipeline includes muscle parameters such as grip strength. At the present the phenotyping and data analysis is still ongoing.

Finally, we would like to thank this reviewer again for his comments and support.